# Microbe-mediated host defence drives the evolution of reduced pathogen virulence

Suzanne A. Ford[1], Damian Kao[1], David Williams[2] & Kayla C. King[1]

Microbes that protect their hosts from pathogens are widespread in nature and are attractive disease control agents. Given that pathogen adaptation to barriers against infection can drive changes in pathogen virulence, 'defensive microbes' may shape disease severity. Here we show that co-evolving a microbe with host-protective properties (*Enterococcus faecalis*) and a pathogen (*Staphylococcus aureus*) within *Caenorhabditis elegans* hosts drives the evolution of reduced pathogen virulence as a by-product of adaptation to the defensive microbe. Using both genomic and phenotypic analyses, we discover that the production of fewer iron-scavenging siderophores by the pathogen reduces the fitness of the defensive microbe and underpins the decline in pathogen virulence. These data show that defensive microbes can shape the evolution of pathogen virulence and that the mechanism of pathogen resistance can determine the direction of virulence evolution.

[1] Department of Zoology, University of Oxford, South Parks Road, Oxford OX1 3PS, UK. [2] Centre for Genomic Research, Institute of Integrative Biology, University of Liverpool, Biosciences Building, Liverpool L69 7ZB UK. Correspondence and requests for materials should be addressed to S.A.F. (email: suzanne.abigail.ford@gmail.com) or to K.C.K. (email: kayla.king@zoo.ox.ac.uk).

Microbes can defend their hosts against pathogenic infection[1]. This phenomenon has been widely observed in nature across plant and animal species, including humans, to provide a strong line of defence beyond the host response[2]. As a result, microbes with protective traits are considered promising disease control agents and alternatives to antibiotics to combat devastating human and crop pathogens[3–6]. Importantly, pathogen adaptation to barriers against infection can drive the evolution of virulence[7], as observed against immune-priming vaccines[8] and competing, co-infecting pathogens[9]. Given that defensive microbes likely impose strong selection on pathogens too[2], they may also play a pivotal role in determining why some infections are more harmful than others. Thus, understanding the effect of defensive microbes on pathogen evolution could ultimately enable us to explain variation in disease severity in nature and to use defensive microbes for optimal disease control.

In this study, we use the term pathogen 'resistance' to describe the pathogen's ability to resist the defensive microbe. We propose that defensive microbes will drive the evolution of pathogen virulence when the mechanism of pathogen resistance to microbe-mediated defence is associated with harm to the host[2,10]. The direction of virulence evolution will depend on the shape of the relationship between resistance and virulence[2,10]. Specifically, we predict that defensive microbes will result in the evolution of increased pathogen virulence if pathogen resistance and virulence are positively associated (Fig. 1a), for example, pathogens could increase in competitive ability against the defensive microbe via a faster growth rate or inducing inflammation, both of which are also detrimental to the host[10–13]. Alternatively, if a trade-off exists between pathogen resistance and virulence traits, for example, resistance is costly and slows pathogen growth rate[14–16], then defensive microbes will drive lower levels of virulence (Fig. 1b). Although the evolution of pathogen resistance to defensive microbes has been theoretically illustrated[10,17] and empirically evidenced[18], studies are few. Our data will thus provide valuable information on the evolution of pathogen resistance in addition to virulence.

Here, we have studied the interactions between *Staphylococcus aureus* and *Enterococcus faecalis* within *Caenorhabditis elegans* hosts which emulate the interactions between a pathogen and a defensive microbe, respectively[19]. Both *E. faecalis* and *S. aureus* co-occur in animal microbiomes and can colonize the gut of *C. elegans*[20]. This nematode species is a model animal system for investigating microbial pathogenesis as well as natural and lab-based host–microbiota associations[21–25]. We used *S. aureus* strain MSSA 476 and *E. faecalis* strain OG1RF, both of which were isolated from humans. Their interaction within a nematode host is therefore novel, allowing us to explore the evolutionary

consequences of defensive symbioses early in their formation. We have previously illustrated that *S. aureus* acts as a virulent pathogen within the nematode host[19]. Furthermore, we have shown that although *E. faecalis* imposes a small cost to host longevity while colonizing alone (un-colonized hosts experience no mortality under otherwise the same conditions[19]), this bacterium provides a net benefit to hosts during *S. aureus* infection[19]. By having context-dependent fitness effects on its host, *E. faecalis* reflects symbionts found in nature. In particular, many defensive microbes have been found to impose costs on their hosts in the absence of the enemy[26–31]. Moreover, we know that *E. faecalis* significantly reduces the virulence of *S. aureus* by producing superoxide anions that decrease pathogen fitness[19]. This mechanism of defence is a form of interference competition, whereby defensive microbes produce toxic compounds that may either kill the pathogen or slow its growth rate[2]. Interference competition is currently the most commonly detected mechanism of microbe-mediated defence in nature[2].

To test the hypothesis that defensive microbes can affect pathogen virulence evolution, we co-passaged *S. aureus* and *E. faecalis* (co-evolution treatment) and independently passaged *S. aureus* (single evolution treatment) within non-evolving, genetically homogeneous *C. elegans* populations for 10 passages. Given that defensive microbes and pathogens likely evolve on similar timescales in nature, co-passaging allowed for the realistic possibility of co-evolutionary interactions to shape pathogen evolution. Throughout our experiments, we assay host mortality after 24 h of exposure to ancestral and evolved pathogen populations as a measure of virulence[32]. We use this measure of virulence for several reasons. Host mortality is a widespread measure in theoretical and experimental investigations on the evolution of virulence across a diversity of host–pathogen systems[9,32–37]. Moreover, host mortality is commonly used in *C. elegans*–pathogen experimental evolution studies[38–40] and our previous work shows its relevance in our system[19]. Specifically, we measure host mortality within 24 h to observe adaptation relevant to the exposure window in the evolution experiment.

We find that experimental co-evolution between a defensive microbe and a pathogen within a host population drives the evolution of reduced pathogen virulence as a by-product of adaptation to the defensive microbe. Using both genomic and phenotypic analyses, we discover that pathogen populations evolved to produce fewer iron-scavenging siderophores and that this change lowered the fitness of defensive microbes and underpinned the decline in pathogen virulence. These data show that microbe-mediated defence can shape pathogen virulence evolution and that the mechanism of pathogen resistance can determine the direction of evolution.

## Results

**Microbe-mediated defence.** Before experimental evolution, we show that *E. faecalis* protects the host from virulent infection after 24 h of simultaneous exposure with *S. aureus* (Fig. 2a, Quasibinomial GLM: F = 41.96, df = 2, P = 5.193e − 09, Supplementary Table 1) by significantly reducing the infection load (Fig. 2b, Welch two-sample *t*-test: t = − 4.5, df = 16.2, P = 0.0003). We find that the benefit *E. faecalis* provides the nematode host (12% reduction in host mortality after 24 h of exposure to the pathogen, Fig. 2a) is much greater than its cost (0.69% mortality after 24 h of exposure to the defensive microbe, Fig. 2a).

Interestingly, we discover that *E. faecalis* reached significantly higher population sizes in the host when it is infected with *S. aureus* (Fig. 2c, Two sample *t*-test: t = − 2.8, df = 24, P = 0.009) suggesting that, although *E. faecalis* harms *S. aureus*, the pathogen conversely alters the environment in a way that

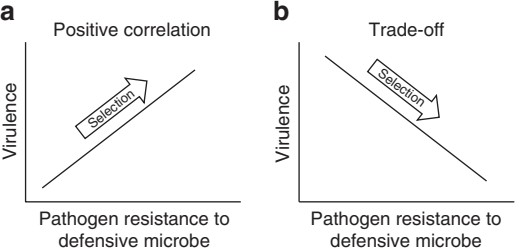

**Figure 1 | Proposed associations between mechanisms of pathogen virulence and resistance to microbe-mediated defence.** (**a**) Pathogen virulence is hypothesized to increase when it correlates positively with the pathogen's ability to resist microbe-mediated defence. (**b**) Pathogen virulence is hypothesized to decrease when traded off against the pathogen's ability to resist microbe-mediated defence.

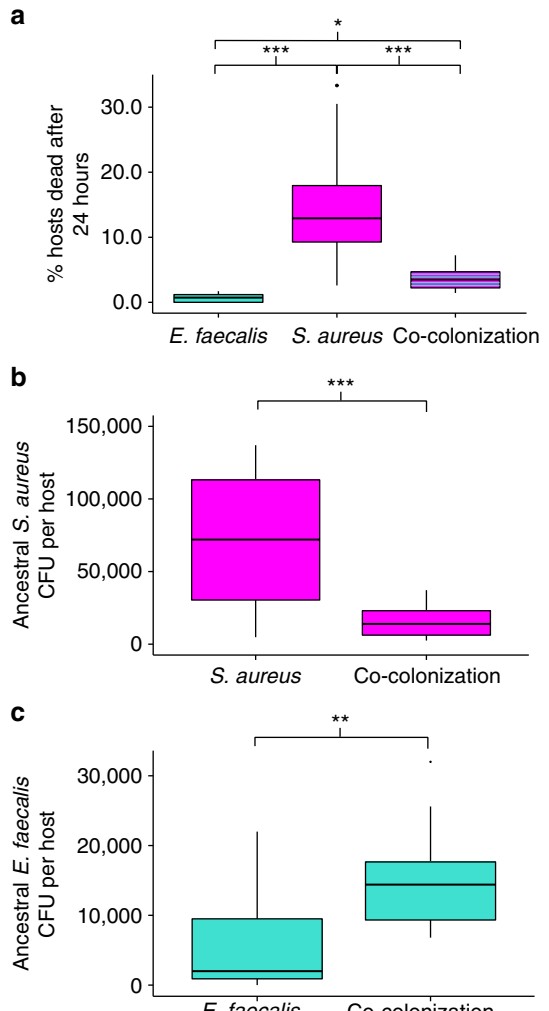

**Figure 2 | Ancestral bacterial virulence and within-host bacterial population sizes.** (**a**) Virulence (% hosts dead after 24 h exposure) of *E. faecalis, S. aureus* and both under co-colonization. Quasibinomial GLM: F = 41.96, df = 2, P = 5.193e − 09. Tukey contrasts: *S. aureus* versus *E. faecalis*: P < 1e − 04; *S. aureus* versus co-colonization: P < 1e − 04; *E. faecalis* versus co-colonization: P = 0.025; Sample size for each treatment: 10 biological replicates. (**b**) Bacterial fitness (colony forming units, CFU per host) of ancestral *S. aureus* under single and co-colonization after 24 h exposure. Welch two sample t-test: t = − 4.5, df = 16.2, P = 0.0003. Sample size for *S. aureus*: 16 biological replicates. Sample size for co-colonization: 18 biological replicates. (**c**) Bacterial fitness (CFU per host) of ancestral *E. faecalis* under single and co-colonization after 24 h exposure. Two sample t-test: t = − 2.8, df = 24, P = 0.009. Sample size for *E faecalis*: 11 biological replicates; Sample size for co-colonization: 15 biological replicates. *P < 0.05, **P < 0.01, ***P < 0.001. Colours indicate the bacterial species being measured: magenta, pathogen (*S. aureus*); turquoise, defensive microbe (*E. faecalis*).

facilitates defensive microbe growth. These data therefore reveal a strong antagonistic relationship between the defensive microbe and the pathogen, whereby one species benefits and the other is harmed.

**Pathogen virulence.** To test whether defensive microbes can affect pathogen virulence evolution, we co-passaged *S. aureus* and *E. faecalis* (co-evolution treatment) and independently passaged *S. aureus* (single evolution treatment) within non-evolving, genetically homogeneous *C. elegans* populations for 10 passages

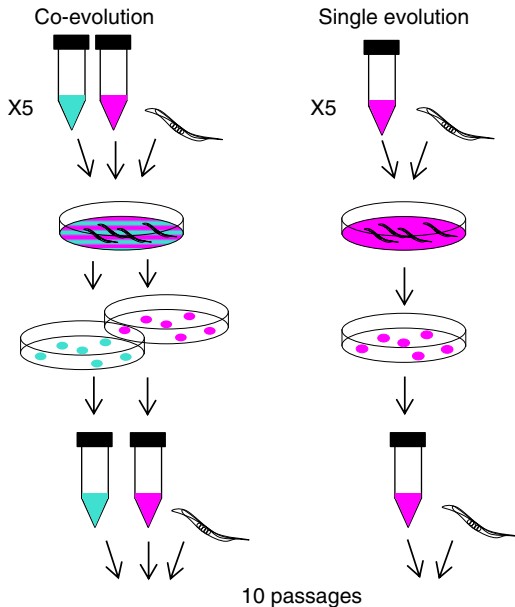

**Figure 3 | Experimental co-evolution.** Experimental co-evolution: *S. aureus* and *E. faecalis* were co-passaged (co-evolution treatment) and *S. aureus* was passaged alone (Single evolution treatment) within non-evolving, genetically homogeneous *C. elegans* host populations. At each generation, the same *C. elegans* host populations were exposed to mixed bacterial lawns of *S. aureus* and *E. faecalis* or single species lawns of *S. aureus*. After 24 h of exposure, 10 colonies of *S. aureus* and/or *E. faecalis* were collected from the pooled guts of 10 dead hosts using selective media and passaged to the next generation. Each treatment had five replicate populations and was conducted for 10 passages. Colours: magenta, pathogen (*S. aureus*); turquoise, defensive microbe (*E. faecalis*).

(Fig. 3). Both treatments consisted of five replicate populations started from the same clone of each bacterial species such that evolution was *de novo*. Every passage, hosts were exposed to bacterial treatments for 24 h, and 10 colonies of *S. aureus* and *E. faecalis* were collected from the pooled gut contents of 10 dead hosts and used to start the next generation (Fig. 3).

We found that experimental co-evolution with the defensive microbe drove reductions in pathogen virulence across all replicate populations, beyond those seen in the single evolution treatment (Fig. 4a, Quasibinomial GLM, F = 25.7, df = 2, P = 4.627e − 05, Supplementary Table 1). This reduction in pathogen virulence correlated with a reduction in *in vitro* pathogen growth rate (Fig. 4b, Pearson's product–moment correlation: t = 3.38, df = 9, P = 0.008, $R^2$ = 0.56), supporting a link between virulence and growth rate, as is commonly assumed[32]. Despite a decrease in growth rate, co-evolved *S. aureus* reached significantly higher infection loads than either the ancestor or single evolution populations in hosts protected by ancestral *E. faecalis* (Fig. 4c, ANOVA, F = 6.55, df = 2, P = 0.012, Supplementary Table 1). This result indicates that the pathogens have adapted to interact with the defensive microbe. Together, these data suggest a trade-off between pathogen virulence and resistance to microbe-mediated defence, supporting our second hypothesis in Fig. 1b.

**Siderophore production.** We proposed that the mechanism underpinning changes in *S. aureus* virulence and growth rate involved the production of siderophores, compounds produced by microbes to bind and uptake iron[41]. In iron-limited host environments, siderophores can affect pathogen growth rate and so contribute to virulence, as evidenced in *S. aureus* infections[42].

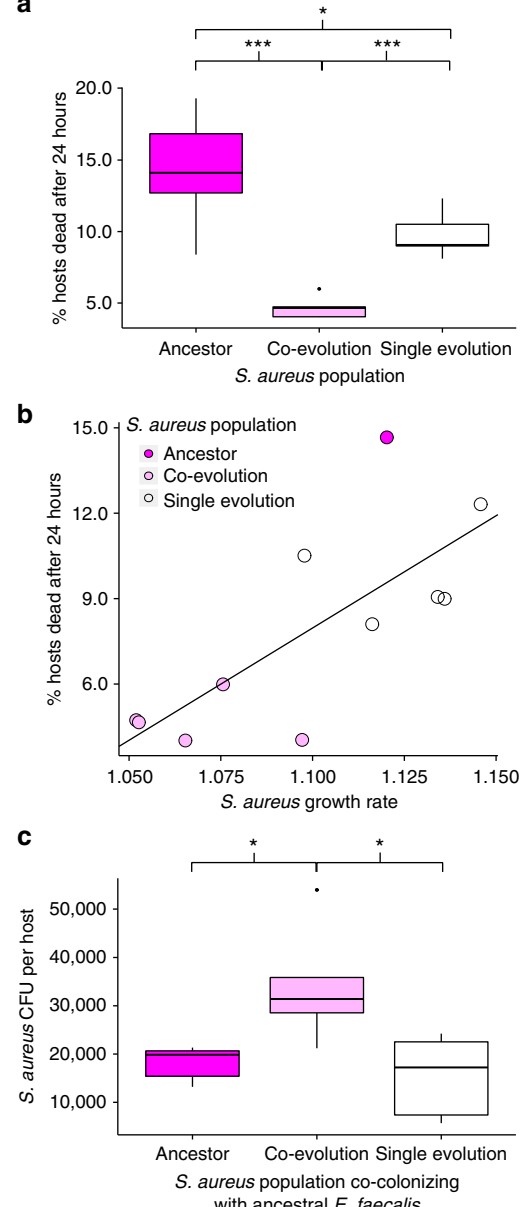

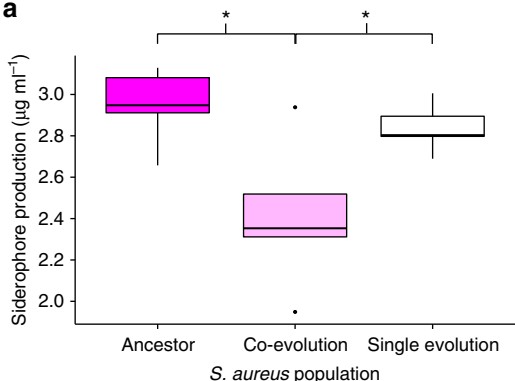

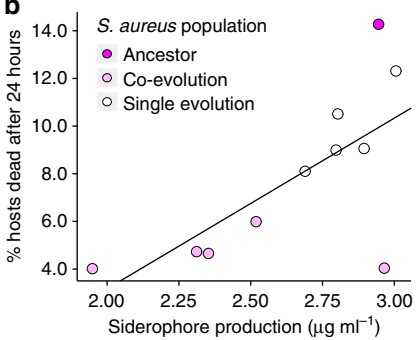

**Figure 4 | Pathogen virulence and fitness.** (**a**) Virulence of ancestral and evolved *S. aureus* populations in *C. elegans* hosts after 24 h of exposure. Quasibinomial GLM: F = 25.7, df = 2, P = 4.627e − 05. Tukey contrasts: co-evolution versus Ancestor: P < 0.001; co-evolution versus Single evolution: P < 0.001; Ancestor versus Single evolution: P = 0.035. Sample size for each treatment: five biological replicates (average of two technical replicates). (**b**) Plot of virulence against the *in vitro* growth rate of ancestral and evolved *S. aureus* populations. Pearson's product-moment correlation: t = 3.38, df = 9, P = 0.008, $R^2$ = 0.56. Sample size for virulence data: as **a** but all ancestral replicates averaged to one biological replicate; Sample size for growth rate data: five biological replicates (average of three technical replicates) for evolved bacteria. One biological replicate for ancestral bacteria (average of eight technical replicates). (**c**) Bacterial fitness (CFU per host) of ancestral and evolved *S. aureus* populations under co-colonization with ancestral *E. faecalis*. ANOVA: F = 6.55, df = 2, P = 0.012. Tukey contrasts: Ancestor versus co-evolution: P = 0.035; Single evolution versus co-evolution: P = 0.015; Single evolution versus Ancestor: P = 0.883. Sample size for each treatment: five biological replicates (one technical replicate). *P < 0.05, **P < 0.01, ***P < 0.001.

**Figure 5 | Siderophore production.** (**a**) Concentration (μg ml$^{-1}$) of siderophores produced by ancestral and evolved *S. aureus* populations *in vitro*. ANOVA: F = 6.681, df = 2, P = 0.01. Tukey contrasts: co-evolution versus ancestor: P = 0.01; single evolution versus ancestor: P = 0.8; single evolution versus co-evolution: P = 0.04. Sample size for each treatment: 5 biological replicates (average of 2 technical replicates). (**b**) Plot of virulence against *in vitro* siderophore production (μg ml$^{-1}$) by ancestral and evolved *S. aureus* populations. Pearson's product-moment correlation: t = 2.8, df = 9, P = 0.02, $R^2$ = 0.5. Sample size for virulence data: as Fig. 4a but all ancestral replicates averaged to 1 biological replicate. *P < 0.05, **P < 0.01, ***P < 0.001.

Crucially, siderophores are also a 'public good' and so can be exploited by neighbouring microbes[41,43]. Theory posits that competition for host resources between unrelated pathogens selects for reduced public good production, and thus reduced virulence[41]. This has been supported by studies of both experimental and natural infections[44,45]. Importantly, when a pathogen is exploited by a defensive microbe, we hypothesize that there would be even stronger selection against public good production, and so virulence, since the exploiting microbe also harms the producer. This extension of current theory would be consistent with our findings that pathogen virulence and growth rate decrease over time in microbe-protected hosts, and would further explain the observed benefit ancestral *S. aureus* confers to *E. faecalis*.

To determine whether the defensive microbe drove a reduction in pathogen virulence as a by-product of selecting for decreased siderophore production, we measured siderophore activity and correlated it with pathogen virulence. By measuring the siderophore concentration produced by *S. aureus* populations *in vitro*, we found that *S. aureus* populations from the co-evolution treatment produced the fewest (Fig. 5a, ANOVA, F = 6.681, df = 2, P = 0.01, Supplementary Table 1). We then discovered that siderophore production was positively correlated with pathogen virulence (Fig. 5b, Pearson's product–moment correlation: t = 2.8, df = 9, P = 0.02, $R^2$ = 0.5). This pattern can be seen both within and between treatments.

**Molecular evolution.** Given that the production of fewer siderophores by co-evolved *S. aureus* was detected in the absence of *E. faecalis*, we concluded that this change was constitutive. To confirm that it was a result of evolutionary change, we conducted whole-genome resequencing of ancestral and *S. aureus* populations at the end of the evolution experiment. No single mutation was shared between any replicate populations, suggesting the independent evolution of the same phenotype multiple times within each treatment (Supplementary Table 2). Importantly, mutations from both evolution treatments can be linked to siderophore production pathways[46] (the putative functions of each mutation is listed in Supplementary Table 2); however SNPs in the co-evolution treatment were present at much higher frequencies compared with those found in the single evolution treatment, consistent with the distinct phenotypes (Fig. 6). Specifically, at least one non-synonymous mutation in each co-evolved pathogen population that had spread to >80% frequency could be linked to siderophore production (Supplementary Table 2). In contrast, only one mutation in one single evolution replicate population reached 50%. This distribution of mutations suggests that defensive microbes imposed strong selection on *S. aureus* populations which resulted in selective sweeps.

**Effect of siderophores on defensive microbe fitness.** To determine whether siderophores were public goods exploitable by the defensive microbe, we measured the *in vitro* growth rate of ancestral *E. faecalis* across a range of exogenous siderophore concentrations. We found that siderophore concentration correlated positively with *E. faecalis* growth rate over the natural range of *S. aureus* siderophore production (Fig. 7a, Pearson's product-moment correlation: $t = 6.1$, $df = 2$, $P = 0.026$, $R^2 = 0.95$). We then found that defensive microbe fitness was directly affected by evolutionary changes in *S. aureus* siderophore production. Here, the fitness benefit *E. faecalis* gained from interacting with the ancestral pathogen (compared with growing independently) disappeared when co-cultured with co-evolved *S. aureus* populations, which produced the fewest siderophores (Fig. 7b, ANOVA, $F = 9.2$, $df = 3$, $P = 0.0009$, Supplementary Table 1).

To test whether changes in siderophore production were sufficient to explain these differences, we measured the effect of surplus iron on *E. faecalis* fitness when co-cultured with ancestral and evolved *S. aureus* populations. Under surplus iron, siderophore production should be downregulated[47]. We found that surplus iron removed the differential effects of *S. aureus* treatment on *E. faecalis* fitness (Fig. 7c, Control ANOVA, $F = 8.1$, $df = 2$, fdr corrected $P = 0.019$; Supplementary Table 1; $+ Fe^{3+}$ ANOVA, $F = 3.05$, $df = 2$, fdr corrected $P = 0.092$). This result confirms public good siderophores as the mechanism underlying pathogen adaptation to microbe-mediated defence in this system.

## Discussion

We sought to establish how within-host interactions between defensive microbes and pathogens would affect the evolution of pathogen virulence. We found that the co-evolution between a defensive microbe and a pathogen drove the evolution of reduced pathogen virulence as a trade-off against pathogen resistance to microbe-mediated defence.

We have previously shown that *S. aureus* is a virulent pathogen within *C. elegans* and that *E. faecalis* acts as a defensive microbe by producing superoxide anions that reduce *S. aureus* fitness[19]. In addition to reducing the fitness of the pathogen, we found here that the defensive microbe could exploit the iron-scavenging siderophores produced by *S. aureus*, gaining a fitness benefit under co-colonization. Together, these data reveal a strong antagonistic interaction between the two species (Fig. 8). By experimentally co-evolving *S. aureus* and *E. faecalis* within nematode populations, we discovered that the pathogens from all replicate populations evolved to produce fewer siderophores. These data correspond with the spread of mutations in genes putatively linked with the siderophore pathway. We confirmed that the decrease in siderophore production resulted in the significant reduction of defensive microbe fitness and was associated with a dramatic increase in pathogen infection load within protected hosts. As a by-product of this adaptation, all co-evolved pathogen populations were significantly less virulent than the ancestral and single evolution populations since decreased siderophore production slowed their growth rate in the absence of the defensive microbe. Given that siderophores are widely used by pathogens[44], reduced siderophore production may be a common adaptation to interactions with members of a host's microbiota and may contribute in explaining the presence of siderophore mutants in pathogen populations in nature.

More generally, our results illustrate that just a single component of a host's microbiota could be a major source of selection acting to shape the virulence of pathogens in nature today. As in our experiment, there are many observations in nature of one defensive microbe species protecting against one enemy species[2]. However, defensive microbe–pathogen interactions may often be more complicated[2], for example complex microbiomes are known to protect species as diverse as bumblebees and humans from infection[3,48], and a single defensive microbe species can defend hosts against multiple enemies[49,50]. We have demonstrated the potential for evolutionary interactions between defensive microbes and pathogens and we have highlighted important implications for virulence. Future research will need to establish how these evolutionary interactions can play out in more complex communities.

The trade-off we observe between pathogen virulence and resistance to microbe-mediated defence is an important consideration for disease control. Host-associated microbes that cause pathogens to make this trade-off, either by similar or different mechanisms to that presented here, could be useful tools to tackle infectious disease. This is not only because they protect hosts in the short-term but they would also select for less virulent pathogens in the long-term. For example, lytic phage that act as defensive microbes by killing pathogenic bacteria can select for decreased pathogen virulence when they infect via bacterial receptors necessary for pathogen growth rate or virulence[16]. By altering these receptors, the pathogen trades-off resistance to phage with virulence mechanisms[16]. In this way, microbe-mediated therapy could provide a more powerful strategy to treat disease than drugs designed to block the production of virulence factors[46] by favouring the evolution of pathogens that are less virulent in the first place. Thus, as an alternative to anti-virulence strategies, which can have unforeseen evolutionary implications[51,52], we could design

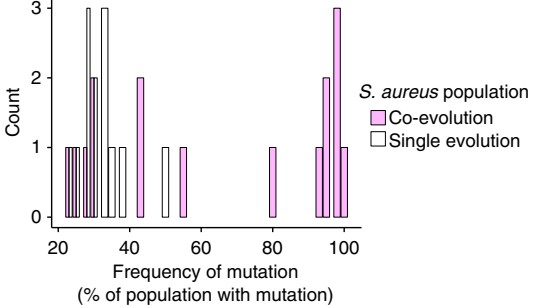

**Figure 6 | Molecular evolution.** Count of non-synonymous mutations in the co-evolution and single evolution replicate populations at a given frequency in the population (the % of 40 randomly sampled clones).

disease control strategies that aim to direct pathogen evolution in a favourable direction.

Microbes that protect hosts from infection can shape the evolution of pathogen virulence. Moreover, it is becoming clear that understanding the mechanisms underlying pathogen resistance to microbe-mediated defence may be crucial in explaining the variation in disease severity in nature and in predicting the direction of virulence evolution under applied disease control.

## Methods

**Nematode host and bacteria.** *C. elegans* is a nematode that constantly interacts with microbes in its natural habitat[53], and it can act as a predator or host for numerous microbial species[22,54,55]. These animals are thus an established model for microbial colonization and pathogenesis[21] and their gut can be co-colonized by multiple pathogens and commensals[38,56–58]. We used the simultaneous hermaphroditic N2 wild-type *C. elegans* strain from the *Caenorhabditis* Genetics Centre (University of Minnesota, Minneapolis, MN)[56]. A genetically homogenous line was generated by selfing a single hermaphrodite for five generations. Populations of these nematodes were frozen in 50% M9 solution and 50% liquid freezing solution in cryotubes at −80 °C (ref. 56). Populations were regularly

resurrected throughout experimentation to prevent the accumulation of *de novo* mutations in host populations. Worms were maintained at 20 °C on nematode growth medium plates seeded with *E. coli* OP50 (*Caenorhabditis* Genetics Centre, University of Minnesota, Minneapolis, MN). *E. coli* OP50 is grown at 30 °C shaking at 200 r.p.m. overnight in LB and 100 µl of this is spread onto nematode growth medium plates and incubated overnight at 30 °C (ref. 56). To ensure clean stocks and to synchronize the life stages of populations for experimentation, worms were treated with bleach (NaClO) and sodium hydroxide (NaOH) solution which kills everything except unhatched worm eggs[56].

We used *S. aureus* strain MSSA 476 (GenBank: BX571857.1), an invasive community-acquired methicillin-susceptible isolate and *E. faecalis* strain OG1RF (GenBank: CP002621.1), both isolated from humans. Both isolates were sourced from the University of Liverpool. A single ancestral population of each species was grown from a single colony overnight in 6 ml Todd Hewitt Broth (THB) shaking at 200 r.p.m. at 30 °C. Bacteria were frozen in a 1:1 ratio of sample to 50% glycerol solution in cryotubes at −80 °C.

**Experimental co-evolution.** The co-evolution experiment consisted of two treatments: (i) *S. aureus* and *E. faecalis* were co-passaged under co-colonization (co-evolution treatment), and (ii) *S. aureus* evolution was passaged on its own (single evolution treatment) within *C. elegans* hosts (Fig. 3a). Each treatment had five replicate populations and was conducted for 10 passages. Every passage, bacteria were cultured overnight in 6 ml THB shaking 200 r.p.m. at 30 °C. They were then standardized to an OD600 reading of 1.00, and 120 µl of each species was spread onto Tryptic Soy Broth (TSB) agar plates and kept overnight at 30 °C

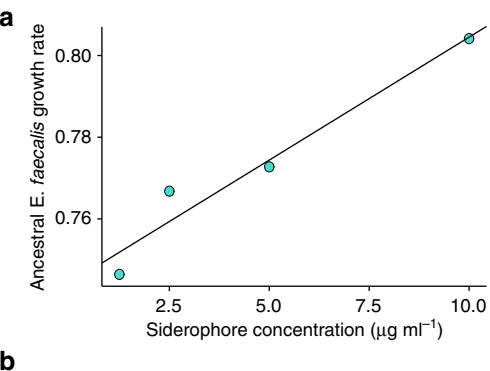

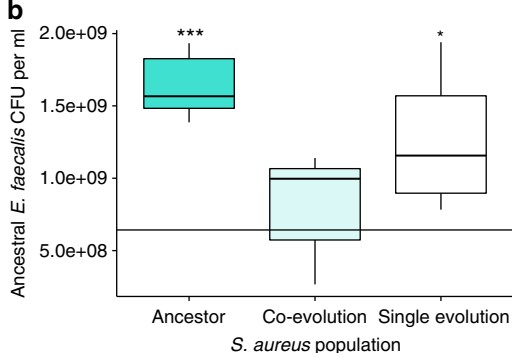

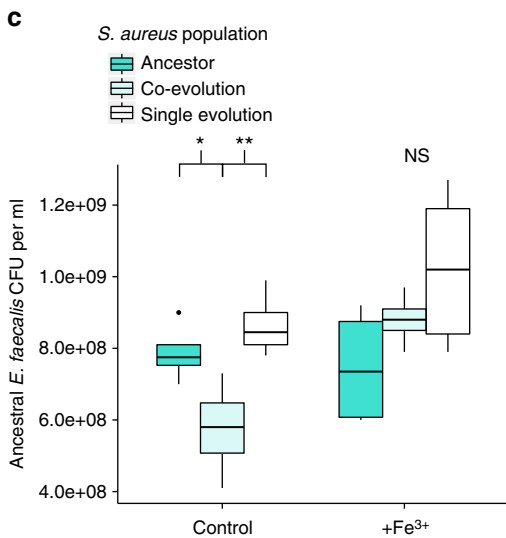

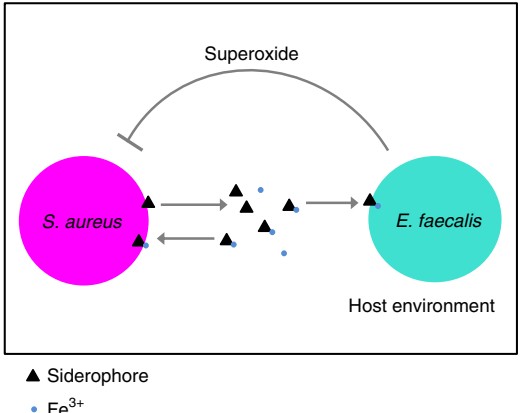

**Figure 8 | Model of *S. aureus*(pathogen)—*E. faecalis*(defensive microbe) interactions within the nematode host.** *S. aureus* produces iron-scavenging siderophores that *E. faecalis* can exploit. In turn *E. faecalis* produces superoxide anions that suppress pathogen growth[19].

**Figure 7 | Effect of siderophores on defensive microbe fitness.**
(**a**) Exogenous siderophore concentration (µg ml⁻¹) against growth rate of ancestral *E. faecalis in vitro*. Pearson's product-moment correlation: $t = 6.1$, $df = 2$, $P = 0.026$, $R^2 = 0.95$. Sample size: one biological replicate for each concentration (average of two technical replicates). (**b**) Fitness of ancestral *E. faecalis* (CFU ml⁻¹) after 24 h growth *in vitro* with ancestral or evolved *S. aureus*. The black line indicates the average fitness of *E. faecalis* in the absence of *S. aureus*. Statistical comparisons are made between this black line and each treatment. ANOVA: $F = 9.2$, $df = 3$, $P = 0.0009$. Dunnett contrasts: co-evolution *S. aureus* and ancestral *E. faecalis* versus ancestral *E. faecalis*: $P = 0.77$; single evolution *S. aureus* and ancestral *E. faecalis* versus ancestral *E. faecalis*: $P = 0.023$; Ancestor *S. aureus* and ancestral *E. faecalis* versus ancestral *E. faecalis*: $P < 0.001$. Sample size for each treatment: five biological replicates (one technical replicate). (**c**) Fitness of ancestral *E. faecalis* (CFU ml⁻¹) after 24 h growth *in vitro* with ancestral or evolved *S. aureus* with ($+Fe^{3+}$) and without (control) surplus iron. Control ANOVA: $F = 8.1$, $df = 2$, fdr corrected $P = 0.019$. Tukey contrasts: Ancestor versus co-evolution: $P = 0.046$; Single evolution versus co-evolution: $P = 0.009$; single evolution versus ancestor: $P = 0.57$. $+Fe^{3+}$ ANOVA treatment: $F = 3.05$, $df = 2$, fdr corrected $P = 0.092$. Sample size for each treatment of both ANOVAs: five biological replicates (one technical replicate). *$P < 0.05$, **$P < 0.01$, ***$P < 0.001$.

to make the exposure lawns. Co-culture exposure lawns for the co-evolution treatment were made by mixing 120 µl of each species together such that their population sizes were kept the same across treatments. Approximately 1,000 young adult worms from the genetically homogenous stock were placed onto each exposure lawn and incubated at 25 °C for 24 h.

After 24 h, 10 dead worms were picked from each lawn to ensure we took hosts colonized with bacteria. Under co-colonization dead worms did not differ significantly from live worms in the proportion of pathogen collected from the infection (Supplementary Fig. 1; two sample $t$-test; $t = 1.22$, df $= 10$, $P = 0.25$). Worms were considered dead when they did not respond to touch with a platinum wire[20]. Worms were washed to remove external bacteria by moving them between five 5 µl drops of M9 buffer on fresh TSB plates using a platinum wire, an adapted method[20,38]. Worms were then crushed in 20 µl of M9 in a 1 ml Eppendorf tube with a pestle to release internal bacteria. An inoculation loop was used to streak each sample onto selective media (TSB plates with 100 µg ml$^{-1}$ rifampicin were used to isolate $E.$ $faecalis$ from the co-evolution treatment, and Mannitol Salt Agar (MSA) plates isolated $S.$ $aureus$ from both treatments) and the plates were grown overnight at 30 °C. Ten colonies of $S.$ $aureus$ and $E.$ $faecalis$ were picked from each replicate and grown in THB overnight, shaking at 200 r.p.m. at 30 °C, and were used for the next generation of exposures.

**Virulence assays.** Exposure plates were made in the same way as for the evolution experiment (see above). Approximately 150 young adult worms from the stock $C.$ $elegans$ population were placed onto exposure plates and incubated at 25 °C for 24 h. Exposure plates were labelled with a random code so that the treatments were unknown during measurement. The total numbers of worms were counted and the proportion dead calculated as a measure of virulence. Worms were considered dead when they did not respond to touch with a platinum wire[20].

**In vivo bacterial population sizes.** Exposure plates were made in the same way as for the evolution experiment (see above). Approximately 1,000 young adult worms from the stock $C.$ $elegans$ population were placed onto exposure plates and incubated for 24 h at 25 °C. Exposure plates were labelled with a random code so that the treatments were unknown during experimentation. After 24 h of exposure, 3–5 worms were picked from each plate. For Supplementary Fig. 1, live worms were picked for each replicate of the live treatment and dead worms were picked for each replicate of the dead treatment. For all other infection load assays dead worms were picked, reflecting the evolution experiment. Worms were washed to remove external bacteria by moving them between five 5 µl drops of M9 buffer on fresh TSB plates using a platinum wire. Worms were then crushed in 20 µl of M9 in a 1 ml Eppendorf tube with a pestle to release internal bacteria. After crushing, serial dilutions were plated onto selective media and grown overnight at 30 °C. MSA and rifampicin TSB selective media were used to count $S.$ $aureus$ and $E.$ $faecalis$ colony-forming units (CFU's) per worm host, respectively.

**In vitro bacterial growth rate.** Bacterial growth rate was measured in 96-well plates. Bacteria were cultured overnight in 6 ml THB shaking 200 r.p.m. at 30 °C. A volume of 2 µl of bacteria was added to 198 µl THB and kept for 10 h in a plate reader set to shake intermittently at 30 °C. OD630 Readings were taken every 1 min and 15 s with Gen5 software and Curve fitter software was used to calculate growth rates of the bacteria in the exponential growth phase.

**Siderophore production.** Ancestral and evolved $S.$ $aureus$ populations were grown in 6 ml THB in falcon tubes with loose cap lids allowing for unlimited oxygen, and so growth, and were shaken at 200 r.p.m. overnight at 30 °C. Overnights were standardized by OD630 and then filtered to remove bacterial cells. One hundred microlitre of each sample was added to 100 µl reaction mixture from the SideroTech (EmergenBio) kit in a 96-well plate and incubated at room temperature for 5 min. OD630 was then measured in a plate reader at room temperature. The concentration of siderophore was estimated from a reference curve of the relationship between OD630 and desferoxamine siderophore (EmergenBio) concentration (Supplementary Fig. 2). This reference curve was made by measuring the OD630 of samples containing desferoxamine siderophore at known concentrations increasing from 0 to 100 µg ml$^{-1}$. The SideroTech assay works by measuring the changes in OD630 when siderophore removes ferric iron from the reagent complex (composed of a dye, iron and a detergent).

**Siderophores and in vitro E. faecalis growth rate.** The growth rate of ancestral $E.$ $faecalis$ was measured in the presence of increasing desferoxamine (EmergenBio) siderophore concentrations in THB. Bacteria were cultured overnight in 6 ml THB shaking 200 r.p.m. at 30 °C. A volume of 2 µl of overnight ancestral $E.$ $faecalis$ culture was added to a mixture of 160 µl THB and 40 µl siderophore-diluent solution at final siderophore concentrations of 1.25, 2.5, 5 and 10 µg ml$^{-1}$. The OD630 was measured every 1 min and 15 s for 10 h in a plate reader set to shake intermittently at 30 °C. OD630 Readings were taken every 1 min and 15 s with Gen5 software and Curve fitter software was used to calculate growth rates of the bacteria in the exponential growth phase.

**In vitro E. faecalis fitness with and without S. aureus.** Bacteria were cultured overnight in 6 ml THB shaking 200 r.p.m. at 30 °C and standardized to an OD630 of 1.00. A volume of 3 µl of ancestral $E.$ $faecalis$ and either ancestral, co-evolution or single evolution $S.$ $aureus$ populations were added to 194 µl of THB in a 96-well plate. As a control, a volume of 3 µl of ancestral $E.$ $faecalis$ was also added to 194 µl of THB without $S.$ $aureus$. The 96-well plate was shaken at 175 r.p.m. for 24 h at 30 °C. Resulting samples were diluted and plated onto rifampicin TSB selective media to count $E.$ $faecalis$ CFU per ml as a fitness measure. Plates were labelled with a random code so that the treatments were unknown during quantification.

**Surplus iron and in vitro E. faecalis fitness with S. aureus.** THB containing 15 µg ml$^{-1}$ iron (III) sulfate was made by mixing THB and iron (III) sulfate dissolved in $H_2O$. Control THB was made by mixing the same volume of THB and $H_2O$ without iron. Bacteria were cultured overnight in 6 ml THB shaking 200 r.p.m. at 30 °C and standardized to an OD630 of 1.00. A volume of 3 µl of ancestral $E.$ $faecalis$ and either ancestral, co-evolution or single evolution $S.$ $aureus$ populations were added to 194 µl of surplus-iron THB or control THB in a 96-well plate. The 96-well plate was shaken at 175 r.p.m. for 24 h at 30 °C. Resulting samples were diluted and plated onto rifampicin TSB selective media to count $E.$ $faecalis$ CFU per ml. Plates were labelled with a random code so that the treatments were unknown during quantification.

**Genomic extraction.** Forty clones were picked from streaked cultures of $S.$ $aureus$ replicate populations from both evolved treatments and grown independently overnight in 200 µl THB in a 96-well plate shaking at 175 r.p.m. at 30 °C. Subsequently, each set of 40 clones was checked for equal OD630 and pooled in equal volumes. A single clone of ancestral $S.$ $aureus$ was similarly grown.

DNA was extracted from $S.$ $aureus$ by taking 1 ml of culture and centrifuging at 7,500 r.p.m., removing the supernatant and re-suspending the pellet in 160 µl enzymatic lysis buffer (Qiagen), 40 µl lysostaphin (200 µg ml$^{-1}$, Sigma-Aldrich), 40 µl lysozyme (100 mg ml$^{-1}$, Sigma-Aldrich) and 8 µl RNAse A (10 mg ml$^{-1}$, Sigma-Aldrich). This mixture was incubated for 1 h at 37 °C. Twenty-five microlitre proteinase K (Qiagen) was then added and 200 µl Buffer AL (without ethanol) was added and vortexed. This was then incubated at 56 °C for 1 h. Two-hundred microlitre of ethanol was added and vortexed and DNA purification followed DNeasy Blood and Tissue Spin-Column Protocol (Qiagen). The extracted DNA was sequenced using the HiSeq4000 platform with 100 bp paired end at the Wellcome Trust Centre for Human Genetics, Oxford.

**Genomic analysis.** The project accession number at the European Nucleotide Archive for the raw read data is PRJEB13385. Sequenced read data in fastq files were trimmed for the presence of Illumina adapter sequences using Cutadapt version v1.9.1 (revision e960cc1 from github.com/marcelm/cutadapt)[59]. The reads were further trimmed using Sickle version 1.33 (revision f3d6ae3 from github.com/najoshi/sickle)[60] with a minimum window quality score of 25 and retaining only reads longer than 50 bp after trimming. Single reads remaining from pairs were retained.

Reference genomes for short read mapping were obtained from the NCBI Assembly database (www.ncbi.nlm.nih.gov/assembly). Short reads of the DNA extractions from the $S.$ $aureus$ strain MSSA476 isolates were mapped to assembly GCF_000011525.1 using BWA MEM v0.7.12 (revision cc9eef2 from github.com/lh3/bwa)[61] using option $-M$ to flag shorter split hits as secondary ensuring the single best alignment was used. Alignments were manipulated using SAMTools v1.2 (revision ac5b8e7 from github.com/samtools/samtools)[62]. The Genome Analysis Toolkit v3.4 (GATK)[63,64] Indel Realigner module was used to realign reads around putative insertions and deletions after which duplicate reads were identified and removed with Picard v1.135 (https://github.com/broadinstitute/picard/releases/tag/1.135). Single nucleotide polymorphism, insertion and deletion discovery was performed with GATK's Haplotype caller module with sample ploidy $n = 40$ (ref. 65). Default parameter values were used except the priors '—heterozygosity 0.0001 –indel_heterozygosity 0.00001' were used because diversity was expected to be relatively low given the experiment duration (these values are ten-fold lower than the defaults). Owing to the difficulties in distinguishing called variants at low frequencies from noise caused by sequencing errors, only variants at frequencies greater than 20% were included in this analysis.

**Statistical analysis.** Data analysis was carried out in R v 3.2.0 (http://www.r-project.org/). Parametric tests were used for all data which met the required assumptions. These assumptions were checked using the Shapiro test to detect whether data was normally distributed and F-tests to compare the variances of two samples from normal populations. Non-parametric equivalents were used if the data did not meet those assumptions. Outlying data points were detected and removed by using the Dixon test. For statistical analyses on passaged bacteria, each replicate population was treated as an independent biological replicate. Thus, there always existed five biological replicates for evolved bacterial population. Quasibinomial GLMs were used for virulence assays to correct for over-dispersion of data, and Tukey contrasts were used for *post hoc* comparisons. $t$-Tests were used for fitness assays of ancestral bacteria within the host. The Welch two sample $t$-test

was used for assays which did not have equal variance and the independent two-sample *t*-test was used when they did have equal variances.

ANOVAs were used for assays comparing ancestral and evolved *S. aureus* treatments measuring within-host bacterial fitness, siderophore production, growth rate and *in vitro* competition of bacteria. Plots of each ANOVA were checked by eye for model quality. For the analysis of ancestral *E. faecalis* in the presence and absence of ancestral and evolved *S. aureus* (Fig. 4b) we performed Dunnett contrasts between treatments with *S. aureus* and the treatment without *S. aureus* as the control. All remaining ANOVAs used Tukey contrasts for *post hoc* comparisons. For the *in vitro* competition assay where a single ANOVA was used for the treatments with surplus iron and a separate ANOVA for the treatments without surplus iron we corrected the treatment *P* values for multiple comparisons using the fdr correction.

Pearson's product–moment correlation was used to test for correlations between *S. aureus* growth rate and siderophore production against virulence, as well as the correlation between siderophore concentration and *E. faecalis* growth rate. The coefficients and $R^2$ value for the lines of best fit were obtained from a linear regression model. To model the relationship between siderophore concentration and optical density a non-linear curve was fitted by estimating the parameters and standard errors of the parameters as starting estimates. The parameters we estimated included the asymptote, intercept and steepest point in curve, 0.2, $-0.53$ and 0.039, respectively. All these parameters appeared to be significant in explaining the model and were not extraneous.

**Data availability.** Raw read data for the bacterial genomic sequences were deposited in the European Nucleotide Archive under project accession number PRJEB13385. CSV files containing data for each figure are available in the Dryad repository (http://dx.doi.org/10.5061/dryad.cb7ns)[66]. The authors declare that all other data supporting the findings of this study are available within the article and its Supplementary Information files, or from the corresponding authors upon request.

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

## Acknowledgements

We thank colleagues from the University of Oxford—Mike Bonsall, Dylan Dahan, Ashleigh Griffin, Charlotte Rafaluk, Ben Sheldon and Stuart West—as well as Helena Mendes-Soares (Mayo Clinic), Greg Hurst (University of Liverpool) and Hinrich Schulenburg (CAU Kiel) for their comments. We are grateful to Steve Paterson (University of Liverpool) for helpful sequencing advice. We also thank the two anonymous reviewers for their positive comments. S.A.F was supported by DPhil funding from Engineering and Physical Sciences Research Council (EPSRC) and St Johns College, University of Oxford. D.K. was supported by Biotechnology and Biological Sciences Research Council (BBSRC) (BB/K007564/1 awarded to Aziz Aboobaker), and K.C.K acknowledges funding from the Royal Society (RG130545) and Leverhulme Trust (RPG-2015-165).

## Author contributions

S.A.F. and K.C.K. conceived and designed the project; S.A.F. conducted the experiments and analysed the data; S.A.F., D.K. and D.W. conducted the bioinformatics analysis; S.A.F. and K.C.K. wrote the paper.
