## [Peer Review File · Nature Communications]

Reviewers' comments:

Reviewer #1 (Remarks to the Author):

This is an interesting study with interesting results that demonstrates a potential mechanism for the evolution of pathogen virulence, that has not previously been considered to my knowledge. I went back and read the recent 2016 Ford and King Opinion piece in PLOS Pathogens to help preface this study to clarify some of points raised in the manuscript.

The authors demonstrate that with mixed infections with *Enterococcus* (Ef), the pathogen *Staphylococcus* (Sa) has reduced density within the nematode host which also has a higher survival rate. Following coevolution, Sa caused reduced host mortality, exhibited lower growth rate and achieved higher density within the host. They also explored one potential mechanism responsibility for these results. In particular, Ef growth increased with siderophore production and grew less with coevolved Sa which produced fewer siderophores. They conclude that Ef exploits Sa-produced siderophores and then produces superoxide anions that suppress Sa growth.

I found that the paper was clearly written and the data presented in Figs. 1 - 4 were clear and convincing. The methods were fairly standard and clearly presented. The take-home message is novel and important - that defensive microbes (DM) can shape evolution of pathogen virulence and that the mechanism of pathogen resistance can determine direction of virulence evolution.

Overall, I liked the paper and think that it will generate considerable interest, but there were a couple of issues that should be addressed. The most important one in my opinion is that the fitness of uninfected nematode populations was never measured (that I could see) or other measures of nematode fitness beyond mortality in the first 24 hours (what about generation time, fecundity?). In particular, the framework of defensive microbes seems to imply that the DM is not just a slightly less virulent pathogen than the pathogen it is defending against. If that is the case, the host would have highest fitness and be better off not being infected with either microbe. Then the paper becomes essentially about bacterial competition. Is there a reason that fitness of uninfected nematodes was not measured? The authors address this issue themselves in their 2016 PLOS Pathogens paper (last three sentences of Box 2).

I think that the authors could address whether the experimental system with one pathogen and one DM may be unusual and whether perhaps more often there are multiple DM (as in a larger microbiome, faecal transplants, etc.), and DM may be active against multiple pathogens. Would coevolutionary interactions be so clear? I also wondered whether the two microbes were isolated from nematodes or taken off the shelf? Does it make a difference whether this is a coevolved system or a de novo interaction?

In line 101, are the SNPs referred to in the siderophore pathway? Is there any evidence that these changes in genome sequence (increasing frequency of mutations) were related to siderophore production, or could they reflect more general interactions with Ef? In general, the siderophore aspect is interesting but is more correlative and not clearly, mechanistically linked to pathogen and DM fitness. Are there other possible mechanisms/alternative hypotheses that would give the same results on microbial virulence and population size? Also, where do the results on superoxide anion production come from?

Specific comments:

Abstract - Reduced production of „public goods" (this is jargon and may not be widely understood without giving a definition, especially in the abstract).

33-35 - provide some details for this statement. Is there evidence for either evolution of resistance

to DMs and if so, is there evidence for the underlying mechanism.

Line 50 - a strong statement. Is this from Table 1 in F and K 2016 paper?

Line 51 - the word colonization can be interpreted two ways. One is the initial process of infection - does Ef initially infect nematodes better if they are already infected with Sa? Or do they reach higher population density within the host if it is infected with Sa? I think the authors mean the latter but could make this clearer.

Line 89 - „this hypothesis" (referring to line 77-78) - this is a very vague hypothesis. What direction, what mechanism?

Line 129 - infection load in infected hosts - this raises the issue again that host fitness was not measured in isolation of the two microbes. They both could be pathogens of the host and the defensive aspect to the interaction is only in the context of one or two microbes infecting, not microbe-free hosts.

Line 144 - inevitable virulence evolution?

Reviewer #2 (Remarks to the Author):

Summary of the key results:

Protective symbioses are ubiquitous across animal and plant systems, yet we know little about their evolution, nor about their consequences for pathogen evolution. This work uses experimental evolution to test the prediction that defensive symbioses will shape the evolution of pathogen virulence. They find that this is indeed the case. Pathogens evolved in the presence of defensive microbes have lower virulence. The authors tie this specifically to alterations in the production of siderophores. This decrease in siderophores production lessens host exploitation and also reduces the ability of the protective microbe to protect the host because they themselves utilize the pathogen produced siderophores as a public good.

I love this paper! On a Friday afternoon after a long week, the clarity and elegant story made me happy. Considering the pathogen side of host-symbiont-pathogen evolution is novel and extremely interesting.

Originality and interest:

This work is novel and will be of interest across many disciplines: experimental evolution, symbiosis, host-microbe interactions, microbe-mediated treatment design.

Data & methodology: validity of approach, quality of data, quality of presentation

No concerns. One minor suggestion is to use the same color coding in Fig2B and 2D as in 2C.

Appropriate use of statistics and treatment of uncertainties

Yes, though would be good in supplement stats table to present confidence intervals and not just P value.

Conclusions: robustness, validity, reliability

No concerns

Suggested improvements: experiments, data for possible revision

No additional data required.

References: appropriate credit to previous work?

No Concerns

Clarity and context: lucidity of abstract/summary, appropriateness of abstract, introduction and conclusions

Excellent.

Response to Reviewers

Reviewer #1:

This is an interesting study with interesting results that demonstrates a potential mechanism for the evolution of pathogen virulence, that has not previously been considered to my knowledge. I went back and read the recent 2016 Ford and King Opinion piece in PLOS Pathogens to help preface this study to clarify some of points raised in the manuscript.

The authors demonstrate that with mixed infections with *Enterococcus* (Ef), the pathogen *Staphylococcus* (Sa) has reduced density within the nematode host which also has a higher survival rate. Following coevolution, Sa caused reduced host mortality, exhibited lower growth rate and achieved higher density within the host. They also explored one potential mechanism responsibility for these results. In particular, Ef growth increased with siderophore production and grew less with coevolved Sa which produced fewer siderophores. They conclude that Ef exploits Sa-produced siderophores and then produces superoxide anions that suppress Sa growth.

I found that the paper was clearly written and the data presented in Figs. 1 - 4 were clear and convincing. The methods were fairly standard and clearly presented. The take-home message is novel and important - that defensive microbes (DM) can shape evolution of pathogen virulence and that the mechanism of pathogen resistance can determine direction of virulence evolution.

We thank the reviewer for their positive and constructive report. We are pleased that the reviewer thought it was 'novel and important' and that the data were 'clear and convincing'. We believe the points brought up by the reviewer have led to a much improved manuscript. We hope we have satisfied all reviewer and editor concerns.

Overall, I liked the paper and think that it will generate considerable interest, but there were a couple of issues that should be addressed. The most important one in my opinion is that the fitness of uninfected nematode populations was never measured (that I could see) or other measures of nematode fitness beyond mortality in the first 24 hours (what about generation time, fecundity?). In particular, the framework of defensive microbes seems to imply that the DM is not just a slightly less virulent pathogen than the pathogen it is defending against. If that is the case, the host would have highest fitness and be better off not being infected with either microbe. Then the paper becomes essentially about bacterial competition. Is there a reason that fitness of uninfected nematodes was not measured? The authors address this issue themselves in their 2016 PLOS Pathogens paper (last three sentences of Box 2).

*We thank the reviewer for highlighting the cost associated with *E. faecalis* colonisation. The cost to hosts of harbouring *E. faecalis* (average of 0.69% mortality in Fig. 3) is a published, inherent feature of the experimental system - we have previously published that uninfected nematodes experience no mortality over the 24 hour experimental period⁴ and have now made reference to these data within the current manuscript (lines 72-77). These costs are also reflective of most natural defensive microbes in the absence of the pathogen, parasite or parasitoid²⁻⁹. Such costs are likely caused by microbes drawing metabolites and energy from hosts, and often affect host survival rate^{4,7}. The key discriminating factor between a parasite and a costly defensive microbe is that the defensive microbe provides a net benefit to the host during infection (paragraph 3 in Box 2 of our PLOS Pathogens opinion piece¹⁰). In our present experiment, we find that the benefit of *E. faecalis* (a 12% reduction in host mortality after 24 hours of exposure to the pathogen) is greater than its cost (0.69% mortality after 24 hours of exposure to the defensive microbe). We have now described this explicitly within the manuscript and thank the reviewer for pointing this issue out (lines 113-115).*

*We have now explained more fully our logic behind quantifying host mortality after 24 hours as our measure of virulence in this system (lines 94-99). Host mortality was chosen as the measure of pathogen virulence for a number of reasons. Firstly, host mortality is a widespread measure in theoretical and experimental investigations on the evolution of virulence across a diversity of host-pathogen systems¹¹⁻¹⁷. Moreover, it is commonly used in *C. elegans*-pathogen experimental evolution studies¹⁸⁻²⁰. Secondly, our previous work demonstrates the relevance of host mortality in our system¹ and so this study builds on these data. Host mortality is therefore a useful measure for placing our study within the current literature, to increase its relevance and enable comparisons to be made across systems and studies.*

Thirdly, host mortality can be accurately measured in nematode populations under the same conditions as in the evolution experiment. To measure generation time and fecundity, however, nematode hosts would need to be placed under different conditions, e.g. rearing nematodes individually instead of in populations of >100 worms, which could alter the outcome of infection. Finally, we specifically measured host mortality within 24 hours to observe adaptation relevant to the exposure window in the evolution experiment.

I think that the authors could address whether the experimental system with one pathogen and one DM may be unusual and whether perhaps more often there are multiple DM (as in a larger microbiome, faecal transplants, etc.), and DM may be active against multiple pathogens. Would coevolutionary interactions be so clear? I also wondered whether the two microbes were isolated from nematodes or taken off the shelf? Does it make a difference whether this is a coevolved system or a de novo interaction?

These are interesting points which highlight the scope for further work and we thank the reviewer for these suggestions. We now include a paragraph elaborating on these points (lines 214-223).

We now state:

'As in our experiment, there are many observations in nature of one defensive microbe species protecting against one enemy species¹⁰. However, defensive microbe-pathogen interactions may often be more complicated this this¹⁰, for example complex microbiomes are known to protect species as diverse as bumblebees and humans from infection^{21,22}, and single defensive microbe species can defend hosts against multiple enemies^{23,24}. We have clearly demonstrated the potential for evolutionary interactions between defensive microbes and pathogens and highlighted important implications for virulence. Future research will need to establish how these evolutionary interactions can play out in more complex communities.'

*Both microbe species, *S. aureus* strain MSSA 476 and *E. faecalis* strain OG1RF, were isolated from humans and so assumed to have no evolutionary history interacting within the nematode host (lines 257-259). Their interaction within a nematode host is therefore novel. This is an important aspect in the design of our experiment because it allows us to explore the evolutionary consequences of defensive symbioses early in their formation. In response to the reviewer's comments we have now highlighted this point within the main text (lines 69-71).*

In line 101, are the SNPs referred to in the siderophore pathway? Is there any evidence that these changes in genome sequence (increasing frequency of mutations) were related to siderophore production, or could they reflect more general interactions with Ef? In general, the siderophore aspect is interesting but is more correlative and not clearly, mechanistically linked to pathogen and DM fitness. Are there other possible mechanisms/alternative hypotheses that would give the same results on microbial virulence and population size? Also, where do the results on superoxide anion production come from?

We have now clarified our molecular results with respect to reviewer comments. The mutations are listed in Supplementary Table 2 along with their putative function. We have now referenced this table more appropriately within the text and have elaborated upon the spread of mutations specifically linked to the siderophore pathway (line 163-173). We find that at least one non-synonymous mutation in each coevolved pathogen population that had spread to >80% frequency could be linked to siderophore production (Supplementary Table 2). In contrast, only one mutation in one Single evolution replicate population reached 50%.

*While it is possible that other mutations resulted in other interactions between the defensive microbe and the pathogen, the siderophore mechanism we examine is the most parsimonious. For example, it is possible that *S. aureus*, at a cost, evolved to produce something harmful to reduce *E. faecalis* fitness. The de novo evolution of such a trait, however, is unlikely in the time of the experimental evolution experiment. Furthermore, this explanation does not account the reduction in siderophore production observed. Our proposed explanation, however, describes: (i) the benefit *E. faecalis* received from ancestral *S. aureus* but not from coevolved *S. aureus*, and (ii) why pathogen virulence and growth rate reduced, particularly given siderophores are molecules widely studied for their relationship to virulence. Furthermore, although we show that the link between *S. aureus* virulence and siderophore production was correlative, we do demonstrate that the change in siderophore production by the coevolved pathogen populations was sufficient to alter the fitness of the defensive microbe (fig.7a and fig.7c).*

*The mechanism of *E. faecalis*-mediated protection via superoxide production is a published aspect of our system¹. We have now clarified this point in the manuscript (lines 78-82 and lines 196-197).*

Specific comments:

Abstract - Reduced production of „public goods“ (this is jargon and may not be widely understood without giving a definition, especially in the abstract).

We have now changed ‘public goods’ to ‘iron-scavenging siderophores’ (line 25).

33-35 - provide some details for this statement. Is there evidence for either evolution of resistance to DMs and if so, is there evidence for the underlying mechanism.

We have now provided information on our current understanding of the evolution of pathogen resistance to defensive microbes (lines 61-63). Studies are currently few but it is likely that the underlying mechanism depends on the mechanism of microbe-mediated defence.

Line 50 - a strong statement. Is this from Table 1 in F and K 2016 paper?

Yes, our literature review indicated that of all defensive microbes studied, interference competition was the most common mechanism of protection¹⁰. However, we recognise that as a field we have incomplete knowledge of all mechanisms of all defensive microbes and so we agree that this is a strong statement and so potentially misleading. Thank-you for highlighting this and we have now changed it to “currently the most commonly detected mechanism” (line 81-82).

Line 51 - the word colonization can be interpreted two ways. One is the initial process of infection - does Ef initially infect nematodes better if they are already infected with Sa? Or do they reach higher population density within the host if it is infected with Sa? I think the authors mean the latter but could make this clearer.

The reviewer is correct that it is the latter. In order to avoid confusion with associated meanings of the word 'colonisation' we have now changed this to "reaches significantly higher population sizes" (line 116).

Line 89 - „this hypothesis" (referring to line 77-78) - this is a very vague hypothesis. What direction, what mechanism?

We have clarified the specific hypothesis we are testing (lines 144-146; 150-151).

Line 129 - infection load in infected hosts - this raises the issue again that host fitness was not measured in isolation of the two microbes. They both could be pathogens of the host and the defensive aspect to the interaction is only in the context of one or two microbes infecting, not microbe-free hosts.

We have address this point above.

Line 144 - inevitable virulence evolution?

We have now removed "inevitable" from this sentence (line 240).

Reviewer #2:

Summary of the key results:

Protective symbioses are ubiquitous across animal and plant systems, yet we know little about their evolution, nor about their consequences for pathogen evolution. This work uses experimental evolution to test the prediction that defensive symbioses will shape the evolution of pathogen virulence. They find that this is indeed the case. Pathogens evolved in the presence of defensive microbes have lower virulence. The authors tie this specifically to alterations in the production of siderophores. This decrease in siderophores production lessens host exploitation and also reduces the ability of the protective microbe to protect the host because they themselves utilize the pathogen produced siderophores as a public good.

I love this paper! On a Friday afternoon after a long week, the clarity and elegant story made me happy. Considering the pathogen side of host-symbiont-pathogen evolution is novel and extremely interesting.

Originality and interest:

This work is novel and will be of interest across many disciplines: experimental evolution, symbiosis, host-microbe interactions, microbe-mediated treatment design.

We thank the reviewer for their overwhelmingly positive comments. The comments made us equally happy!

Data & methodology: validity of approach, quality of data, quality of presentation

No concerns. One minor suggestion is to use the same color coding in Fig2B and 2D as in 2C.

We have now used consistent colour-coding throughout the document. We have decided to use magenta and turquoise instead of red and green to avoid colour-blind issues as advised by the Nature Communications formatting guidelines. Magenta represents S. aureus traits being measured and turquoise represents E. faecalis traits being measured. In each graph we use different shades of these colours to represent the different treatments of each species.

Appropriate use of statistics and treatment of uncertainties

Yes, though would be good in supplement stats table to present confidence intervals and not just P value.

We have now included confidence intervals in our Supplementary Table when available from the analysis.

Conclusions: robustness, validity, reliability

No concerns

Suggested improvements: experiments, data for possible revision

No additional data required.

References: appropriate credit to previous work?

No Concerns

Clarity and context: lucidity of abstract/summary, appropriateness of abstract, introduction and conclusions

Excellent.

REVIEWERS' COMMENTS:

Reviewer #1 (Remarks to the Author):

I reviewed the initial submission of this manuscript. My major comment then was that no data were provided on fitness of uninfected nematodes (to compare fitness relative to nematodes with the defensive symbiont only, and nematodes with both symbiont and pathogen). This point has been addressed by the authors and in the manuscript, which is improved as a result.

I also had questions about host mortality as the only measure of fitness, the relevance of this simplified study to more complex microbiomes and defensive interactions, and whether the experimental system was naturally occurring in nematodes or constituted by getting microbes from other systems or from culture collections. These issues have also been satisfactorily addressed in the revised manuscript.

My smaller, more specific comments have also been addressed. I also noted the comments of the other reviewer, noting that they did not really have any significant concerns.

Finally, I read the entire revised manuscript again and only had one minor point: in the paragraph starting on line 52 (Introduction), it could be unclear if the term "pathogen resistance" (referring specifically to microbe-mediated defense on line 53 and line 61) is used consistently in that way vs. what could be construed as host resistance to the pathogen, or even pathogen resistance to host traits that deter infection. A short sentence just saying we use "pathogen resistance" to mean would suffice.

Overall, this is an interesting and well-written paper. I think that the revisions addressed my initial concerns and make for an improved paper as a result. It will be of general interest to readers interested in pathogen evolution and microbe-microbe interactions within hosts.